# Autonomous learning of features for control: Experiments with embodied and situated agents

**Nicola Milano** *, **Stefano Nolfi**

Institute of Cognitive Science and Technologies, National Research Council (CNR-ISTC), Roma, Italy

* nicola.milano@istc.cnr.it

## Abstract

The efficacy of evolutionary or reinforcement learning algorithms for continuous control optimization can be enhanced by including an additional neural network dedicated to features extraction trained through self-supervision. In this paper we introduce a method that permits to continue the training of the features extracting network during the training of the control network. We demonstrate that the parallel training of the two networks is crucial in the case of agents that operate on the basis of egocentric observations and that the extraction of features provides an advantage also in problems that do not benefit from dimensionality reduction. Finally, we compare different feature extracting methods and we show that sequence-to-sequence learning outperforms the alternative methods considered in previous studies.

## 1. Introduction

The discovery of expressive features constitutes a critical pre-requisite for the development of effective control policies. In the case of robots, feature learning can be especially challenging as it involves the generation of features describing the state of the agent and of the environment from raw and noisy sensor measurements that provide only part of the required information.

Recent works in evolutionary [1, 2] and reinforcement learning methods [3, 4] demonstrated how adaptive robots can successfully learn effective policies directly from observation on the basis of reward signals that rate only the performance of the agent. In other words, the adaptive robots can discover the features required to realize effective control policies automatically without the need of additional dedicated mechanisms or processes. This approach is usually referred as end-to-end learning. On the other hand, extended methods incorporating mechanisms or processes dedicated to the extraction of useful features can potentially speed-up learning and achieve better performance.

Feature learning is a general domain which aims to extract features that can be used to characterize data. Recently, this area achieved remarkably results in the context of neural network learning for classification and regression problems. For example, feature learning with deep convolutional neural networks constitutes an excellent solution for image classification problems. Autonomous learning of features for control is a special area of feature learning in which the input vectors (*observations*) are influenced by output vectors (*actions*) and in which

**Competing interests:** The authors have declared that no competing interests exist.

observations and actions vary during the course of the learning process [5, 6]. The term autonomous refers to the fact that the features are extracted through a self-supervised learning process, i.e. a learning process in which the supervision is provided directly by the data available as input.

The methods that can be used to learn features include auto-encoders, forward models, and cognitive biases or priors. Auto-encoders are trained to encode observations in more compact intermediate vectors that are then decoded to regenerate the original observation vectors. Forward models are trained to predict the next observation on the basis of the current observation and of the action that the agent is going to perform. The features extracted by auto-encoders or forward models can then be provided as input to the control network that is trained through evolutionary or reinforcement learning algorithms to maximize a task-dependent reward function. Cognitive biases or priors are constraints that promote the development of features characterized by certain desired properties in standard end-to-end learning, for example features fluctuating slowly over time [7–9]. Constraint of these types can be introduced by using cost functions that encourage the generation of features with the desired qualities.

A first objective of feature learning consists in generating compact representation of high dimensional sensory data which, in the case of robotic problems, might include 50 or more joint angles and velocities, hundreds of sensors encoding tactile information, and thousands or millions of pixels forming visual images. A second objective consists in generating useful features, i.e. features that are more informative than the observation states from which they are extracted.

We focus on the second objective. More specifically, we analyze whether the features extracted through self-supervised learning facilitate the development of effective solutions in embodied and situated agents trained through an evolutionary algorithm. Unlike previous related works, we consider experimental scenarios in which the robots have access to egocentric perceptual information. Moreover, we consider experimental scenarios that do not involve high-dimensional observations and, consequently, do not benefit from dimensionality reduction.

We introduce a method that permits to continue the training of the feature extracting network during the training of the control network and we validate the method in combination with different feature-extracting models, including the sequence-to-sequence method [10–12] that has not been used in previous works.

Our results demonstrate that continuing the training of the feature extracting network leads to much better results than the technique used in previous works in which the training of the network is completed before the training of the control network. The comparison of the results obtained by extracting the feature through an auto-encoder, and auto-encoder and a forward model, and a sequence to sequence model indicates that all methods provide an advantage with respect to end-to-end learning and that the best performance are achieved through the sequence-to-sequence model.

## 2. Novelty and relation with related works

Previous related works considered scenarios in which the agent can observe itself and the environment from an external perspective. In particular, [13] considered the case of a racing slot car that can observe itself and the race track through a camera located above. The car is trained through reinforcement learning to move as fast as possible without crashing by receiving as input a low-dimensional representation of the scene extracted from a deep autoencoder neural network. This is realized in three phases. The first phase is used to generate the training set for the auto-encoder network that is created by collecting images from the camera at a constant

frame rate while the car moves at a low constant speed on the track. The second phase is used to train the autoencoder network to learn an abstract compressed representation of the images (i.e. to compress the observations in a vector $z$ that is then used to regenerate the original observations). Finally, the third phase is used to train the control network to drive the car. The control network receives as input the vector $z$, extracted from the observations by the auto-associative network, and produces as output the desired speed of the wheels of the car. A similar approach has been used to train an agent to swing a pendulum on the basis of visual information collected from a side view [14]. The results reported by the authors demonstrate that the usage of the auto-encoder network permits to speed-up learning with respect to an end-to-end approach.

More recently [15], proposed a method that combines an auto-encoder network, a forward-model network, and a control network. The model has been applied to the CarRacing-v0 problem [16] that consists in driving a car on a race track in simulation on the basis of images collected by a camera located above the track. Also in this case, the first phase is dedicated to the collection of the training set that is generated by collecting the observations $o$ experienced by the car and the actions $a$ performed by the car during 10,000 episodes in which the car moves by selecting random actions. The second phase is used to train the auto-encoder network. The training enables the auto-encoder to extract an abstract compressed representation $z$ of observations. During the third phase, the forward-model network is trained to predict $z_{t+1}$ on the basis $z_t$ and $a_t$. This enables the forward model to compress the observations of the agent over time in a vector $h$. Both the auto-encoder and the forward-model networks are trained on the basis of the training set collected in the first phase. Finally, in the fourth phase, the control network is trained to drive the car so to maximize the cumulative reward. The control network receives as input the $z$ and $h$ vectors extracted by the auto-associative and forward models and is trained through the Covariance-Matrix Adaptation Evolution Strategy (CMA-ES) [17]. As shown by the authors, this method outperforms a standard end-to-end training method in which the control network receives in input the observations.

In this work we investigate whether autonomous feature learning is beneficial also in problems in which the agents receive egocentric observations, i.e. in problems in which the sensors of the agents are located on the agents' body. The usage of allocentric observations, e.g. images collected from a camera located outside the agent, implies the necessity to transform the information contained in the observations to the agent's perspective, i.e. to the perspective that is relevant to decide the actions that the agent should perform. Consequently, the advantage of feature-extraction reported in previous studies can be ascribed, at least in part, to the need to perform this transformation. To verify whether the advantage of feature extraction is limited to problems that involve allocentric/egocentric transformation or not we thus selected problems that operate on the basis of egocentric information. More specifically we choose the BulletWalker2D, the BulletHalfcheetah [18], the BipedalWalkerHardcore [19], and the MIT Racecar problem (https://racecar.mit.edu/ [18]), i.e. a set of widely used problem that are often used to benchmark alternative algorithms.

These problems are also qualitatively different from previous studied for what concerns the size of the observation vector that is relatively compact and much smaller than in the problems reviewed above. This enable us to verify whether feature extraction is advantageous in general or whether it provides an advantage in problems that benefit from dimensionality reduction only.

Beside of that, the main novelty of this study consists in the prolonged training of the feature-extracting network and in the utilization of the sequence-to-sequence learning model. As we will see, both aspects permit to significantly improve the advantage that can be gained from the features extracting network/s.

## 3. Experimental setting

The Walker2DBullet (Fig 1, top-left) and HalfcheetahBullet (Fig 1, top-right) problems involve simulated robots with different morphologies rewarded for the ability to walk on a flat terrain [18]. The bullet version of these problems constitutes a free and more realistic implementation of the original MuJoCo problems [20]. The robots have 6 actuated joints. The observation vector includes the position and orientation of the of the robot, the angular position and velocity of the actuated joints, and the contact sensors located on the feet. The initial posture of the robot varies randomly within limits. The evaluation episodes are terminated after 1000 steps or, prematurely, when the height of the torso diminishes over a threshold. The robots are rewarded on the basis of their velocity toward the target destination in m/s. In the case of the HalfcheetahBullet, the reward function includes an additional component that punishes the agent with -0.1 in each step for each joint currently located on one of the two limits. These rewards functions, optimized for evolutionary strategies, include fewer components than those optimized for reinforcement learning (see [2]).

The BipedalWalkerHardcore (Fig 1, bottom-left) is another walker problem that involves a two-legged 2D agent with 6 actuated joints situated in an environment including obstacles

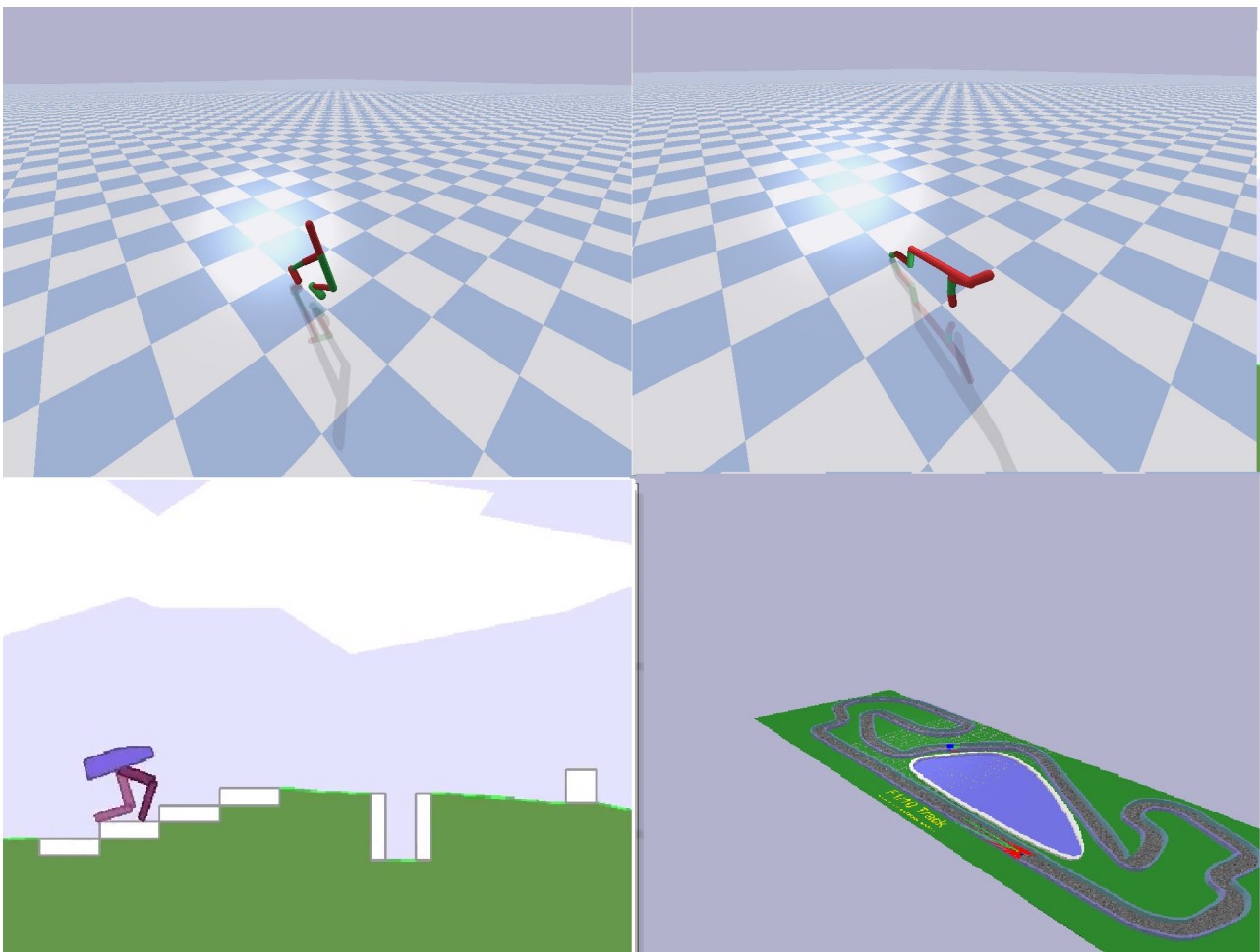

**Fig 1.** Illustration of the Walker2DBullet (top-left), HalfcheetahBullet (top-right), BipedalWalker (bottom-left), and MIT racecar (bottom-right) problems.

constituted by ladders, stumps, and pitfalls. The agent is rewarded for moving forward and is penalized proportionally to the torque used. The agent's observation includes the horizontal and vertical speed of the hull, the angular speed and velocity of the hull, the position and speed of the joints, the state of the contact sensors placed on the terminal segments of the legs, and 10 proximity measures collected by a lidar rangefinder.

The MIT racecar problem involves a simulated version of the MIT racecar located on a race track (Fig 1, bottom-right). The car has 2 actuators that control acceleration and steering. The observation includes 30 lidar proximity measures distributed over a range of 270 degrees on the frontal side of the car. The racetrack is divided in 110 consecutive virtual sections and the car is rewarded with 1 point every time it moves in the following sector. The evaluation episodes are terminated after 10,000 steps or, prematurely, if the car remains stuck for 100 consecutive steps.

The size of the observation and action vectors are [22, 26, 22, 30] and [6, 6, 6, 2] in the case of the Walker2dBullet, HafcheetahBullet, BipedalWalkerHardcore, and MIT racecar problems, respectively. The control network, that determines the state of the actuators, is constituted by a feed-forward neural network with an internal layer composed of 64 internal neurons. The internal and motor neurons are updated with hyperbolic tangent and linear activation functions, respectively. The number of the output neurons corresponds to the length of the corresponding action vector. The number and type of sensory neurons vary in different experimental conditions (see below). The connection weights and the biases of the control network are trained through the Open-AI evolutionary strategy [1], i.e. one of the most effective evolutionary methods for continuous problem optimization [2]. Agents are evaluated for 1 episode in the case of the Walker2DBullet, HalfCheetahBullet, BipedalWalkerHardcore, and MIT racecar and for 5 episodes in the case of the BipedalWalkerHardcore. The usage of a greater number of episodes in the last case is due to the fact that the variability of the environmental conditions is greater. Episodes last 1,000 steps in the case of the Walker2DBullet, Half-CheetahBullet, and BipedalWalkerHardcore problems, and 10,000 steps in the case of the MIT racecar. The duration is longer in the last case since the car need a significant amount of time to complete a lap of the race track. The training of the control network is continued for $20 * 10^7$ steps in the case of the BipedalWalkerHardcore problem, which requires more time than the other problems to achieve good performance, and for $5 * 10^7$ steps in the other cases. To estimate the performance of the evolving agents more accurately, we post-evaluate the best agent of each generation for 20 episodes, in the case of the BipedalWalkerHardcore, and for 3 episodes in the other cases.

We considered five experimental conditions (Fig 2) and two modalities.

In the end-to-end (**EtE**) experimental condition the control network receives as input directly the observation.

In the autoencoder (**AE**) experimental condition the control network receives as input the features extracted from the observations by the autoencoder, i.e. a feed-forward neural network with 1 internal layer composed of 50 neurons.

In the autoencoder forward-model experimental condition (**AE-FM**) the control network receives as input the features extracted by an auto-encoder network and by the forward model network. The former is constituted by a feed-forward network with 1 internal layer formed by 50 neurons. The latter is constituted by a LSTM network [21] with 1 internal layer formed by 50 units. As in [15], the forward-model network is trained to predict the internal state of the autoencoder $z$ at time $t_{+1}$ on the basis of the internal state of the autoencoder at time $t$ and of the action that the agent is going to perform.

In the sequence-to-sequence experimental condition (**StS**) the control network receives as input the features extracted by a sequence-to-sequence network that is trained to compress the last 5 observations into an internal vector $h$ that is used to regenerate the same observations.

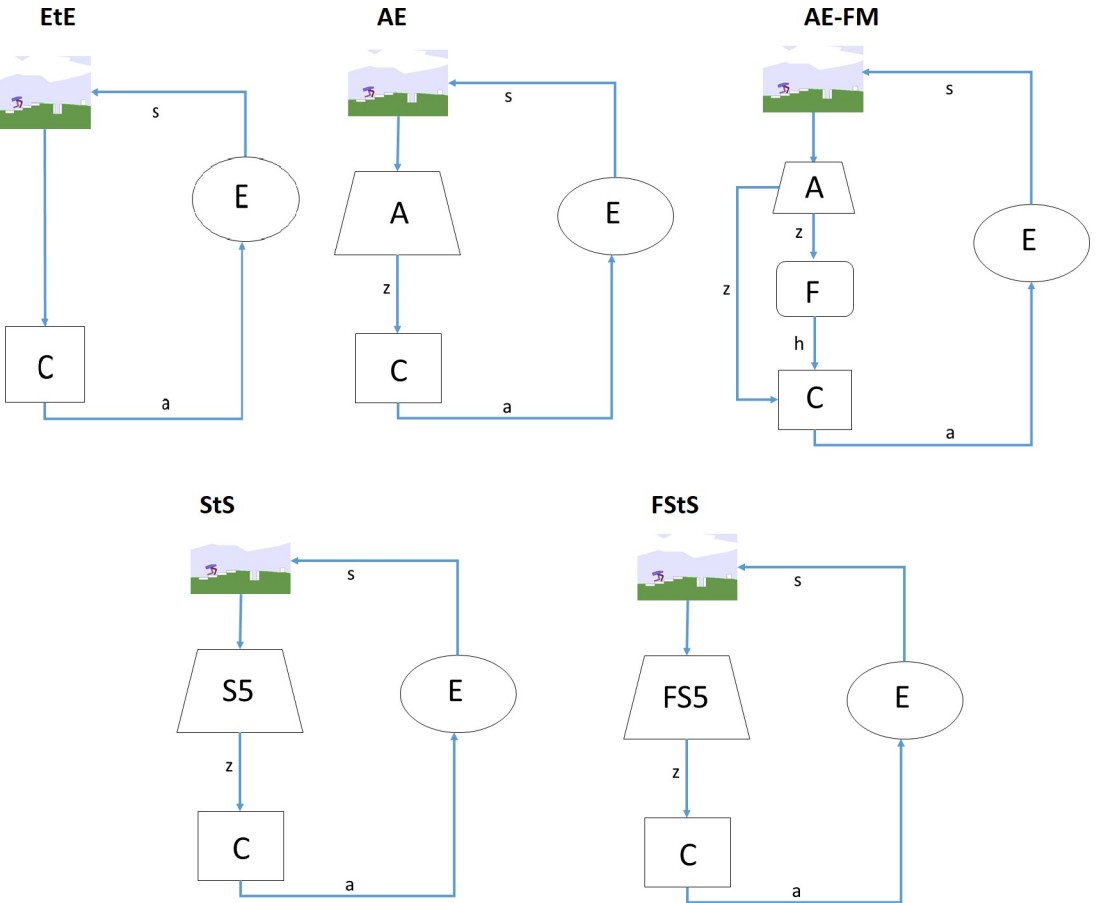

**Fig 2. Schematization of the 5 experimental conditions (EtE, AE, AE-FM, StS, and FStS).** The picture at the top of each image indicate the observation (e.g. the angle and velocity of the agent's joint, the proximity measures ext.). The action vectors are indicated with the letter $a$. The vector of features extracted from the auto-associative or by the sequence-to-sequence network are indicated with the letter $z$. The vector of features extracted by the forward model is indicated with the letter $h$. A indicates an auto-associative network that receives as input the observation at time $t$ and produce as output the same observation at time $t$. F indicates a forward model network that receives as input the vector $z$, extracted from the auto-associative network at time $t$, and the action vector $a$ at time $t$ and produces as output the vector $z$ at time $t_{+1}$. S5 indicates a sequence-to-sequence associative network that receives as input the observations at time $[t_{-4}, t]$ and produces as output the observation at time $[t_{-4}, t]$. FS5 indicates a sequence-to-sequence network that receives as input the observation at time $[t_{-4}, t]$ and produces as output the observations at time $[t_{-3}, t_{+1}]$. C indicates the control network that receives as input the observation (in the case of the EtE condition), or the features vector $z$ (in the case of the AE, StS, and FStS conditions), or the features vectors $z$ and $h$ (in the case of the AE-FM condition). C produces as output the action vector that determines the movement of the agent's joints or wheels and consequently the state (s) of the environment (E) at time $t_{+1}$. The new state of the environment determines the observation at time $t_{+1}$.

Finally, in the forward sequence-to-sequence experimental condition (**FStS**) the control network receives as input the feature extracted from a sequence to sequence network that is trained to compress the last 5 observations into an internal vector $h$ that is used to regenerate the last 4 observation and to predict the following future observation.

In the **pre-training** modality, the training set of the feature-extracting network/s is collected during 1000 episodes in which the agents perform random actions. The training of the feature-extracting network/s is continued for 500 epochs and is completed before the training of the control network. In the **continuous training** modality, the training set is updated every generation and the training of the feature-extracting network/s is continued during the training of the control network. The training set is initially collected as in the pre-training modality

but is updated every generation by replacing the oldest 1% of the data with the observations collected during the evaluation of the best agent of the current generation. The feature extracting network/s are trained for 10 epochs every generation.

The experiments performed can be replicated by using the software available from https://github.com/milnico/features-extraction

# 4. Results

Fig 3 shows the performance obtained in the five different experimental conditions in the case of the Walker2DBullet problem (Fig 3). The top figure shows the results obtained in the pre-

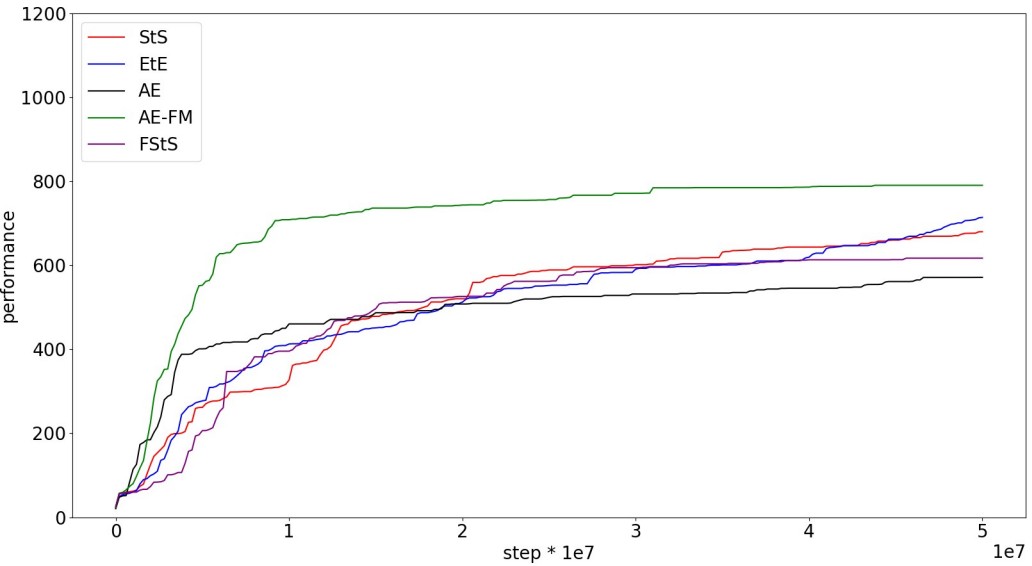

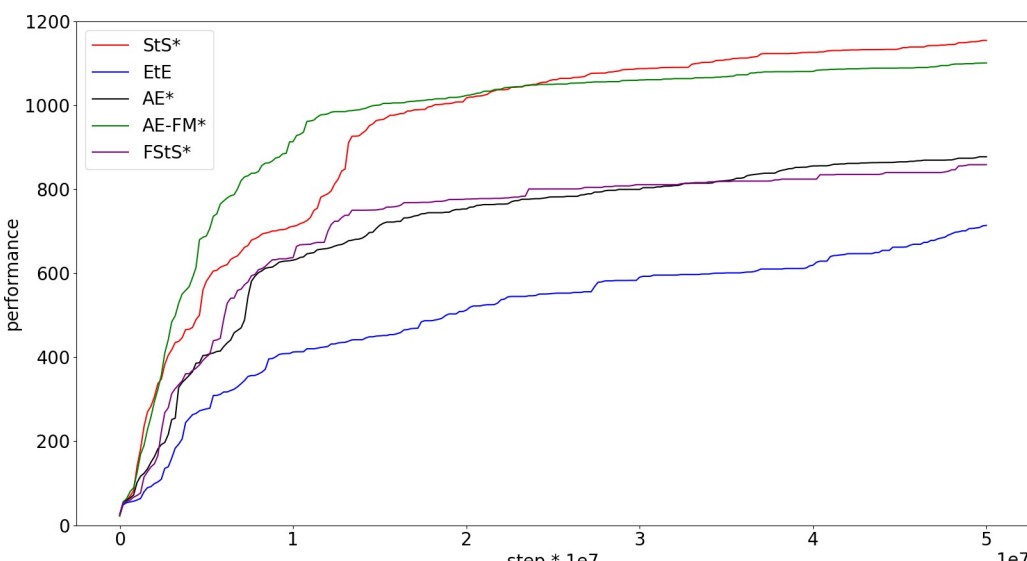

**Fig 3. Performance during the training process in the case of the Walker2DBullet problem.** Performance refer to the average fitness of the best agent achieved so far. Data computed by post-evaluating the best individual of each generation for 3 episodes. Each curve shows the average result of 10 replications. The top and the bottom figures show the results obtained in the experiments performed in the pre-training and continuous training modalities, respectively.

training modality, in which the training of the feature extracting network/s is completed before the training of the control network. The bottom figure, instead, shows the results obtained in the continuous training modality in which the training of the feature-extracting network/s is continued during the training of the control network and in which the training set is updated with observations collected by agents acting on the basis of their current control network.

The experiments performed in the continuous training modalities outperform the experiments performed in the pre-training modality in all cases (Fig 3 top and bottom, Mann-Whitney U test p-value < 0.01 in all conditions). Data obtained by comparing the best agents obtained at the end of the training process. This can be explained by considering that the observations experienced by agents performing random actions differ significantly from the observations experienced by agents trained to maximize their fitness. In other words, the features extracted from observation experienced after the execution of random actions do not necessarily capture the regularities that characterized the observations experienced by evolving agents. This hypothesis is supported by the fact the offset between the actual and desired outputs produced by the autoencoder, forward-model, sequence to sequence, and forward sequence to sequence networks increase considerably during the training of the control network in the case of the pre-training modality. On the contrary, the offset remains low in the experiments performed in the continuous training modality (Fig 4, base versus * conditions).

All feature-extracting experimental conditions outperform the EtE experimental condition in the case of the continuous training modality (Mann-Whitney U test p-value < 0.01 in all conditions). The best performance are obtained by the sequence to sequence condition (StS*) that produces significantly better performance than all the other conditions (Mann-Whitney U test p-value < 0.01).

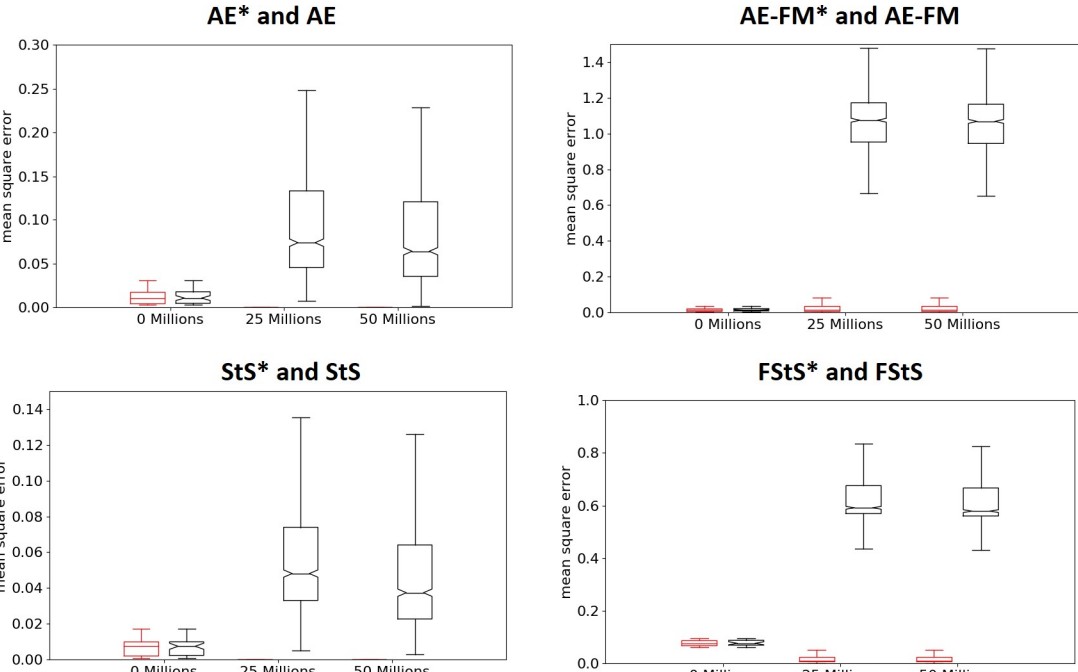

**Fig 4. Mean-squared error produced by the features-extracting networks before the training of the control network (0 steps) and during the training of the control network (after 25 and 50 $\ast$ $10^6$ steps).** Results of the experiments performed in the AE, AE-FM, StS, and FStS experimental conditions. The boxplots shown in black and red shows the data of the experiments performed in the pre-trained and continuous training modalities, respectively. In the case of AE* and StS* conditions, some of the red boxplots are not visible because the distribution of the offset is close to 0.0.

Fig 5 shows the performance in of StS* and the EtE experimental conditions in the case of the other three problems considered. We focused on the StS* method since it achieved the best performance in the case of Walker2DBulletProblem. The StS* method outperforms EtE method in all case (Mann-Whitney U test p-value < 0.01) with the exception of the HalfcheetahBullet in which the performance in the two experimental conditions do not differ statistically (Mann-Whitney U test p-value >0.01). Data obtained by comparing the best agents obtained at the end of the training process.

## 5. Discussion

The efficacy of evolutionary or reinforcement learning algorithms can be improved by combining the control network with one or more neural networks trained to extract abstract features from observations through self-supervised methods. Indeed, previous works demonstrated how combined models of this type can speed-up learning and/or achieve better performance also in continuous problems domains. In particular, the research reported in [13–15] demonstrated how the addition of feature-extraction network/s is beneficial, at least in the case of problems that can benefit from dimensionality reduction and that operate on the basis of allocentric observations.

In this paper we introduced a method that permits to continue the training of the feature extracting network/s during the training of the control network, we benchmarked alternative feature extracting models including the sequence-to-sequence model that has not been used in previous works, and we validated feature extracting methods on problems that operate with egocentric perceptual information.

Our results demonstrate that the continuous training modality introduced in this paper produces much better results than the pre-training modality used in previous works. The necessity to continue the training of the feature-extracting network/s during the training of the control network can be explained by the fact that the problems considered involve agents operating on the basis of egocentric information while the problems considered in the other studies referred above involve agents operating on the basis of allocentric information, i.e. on the basis of a camera detached from the agent that observe the agent and the environment. Indeed, the difference between the observations experienced by producing random action with respect to actions selected to maximize the expected reward is greater for agents that operate on the basis of egocentric information than for agents that operate on the basis of allocentric information.

Moreover, our results show that the sequence-to-sequence model outperforms the other feature extracting methods used in previous works. The sequence-to-sequence method was not considered in previous related studies. Its efficacy can be explained by the fact that it permits to extract features characterizing variations over longer time spans than forward models.

The utilization of problems that involve agents operating on the basis of egocentric information, instead of allocentric information as in previous studies, demonstrates that feature extraction can be advantageous in general terms, irrespectively from the necessity to perform a perspective transformation. Moreover, the utilization of problems that involve relatively compact observation vectors demonstrates that the advantages of features extraction goes further dimensionality reduction.

Future studies should verify the scalability of this method to more complex problems, e.g. to agents possessing more DOFs situated in a more complex environments. The advantage which can be gained from feature extraction can be higher in more complex setting. On the other hand, the problem of extracting suitable features can become more challenging in more complex settings.

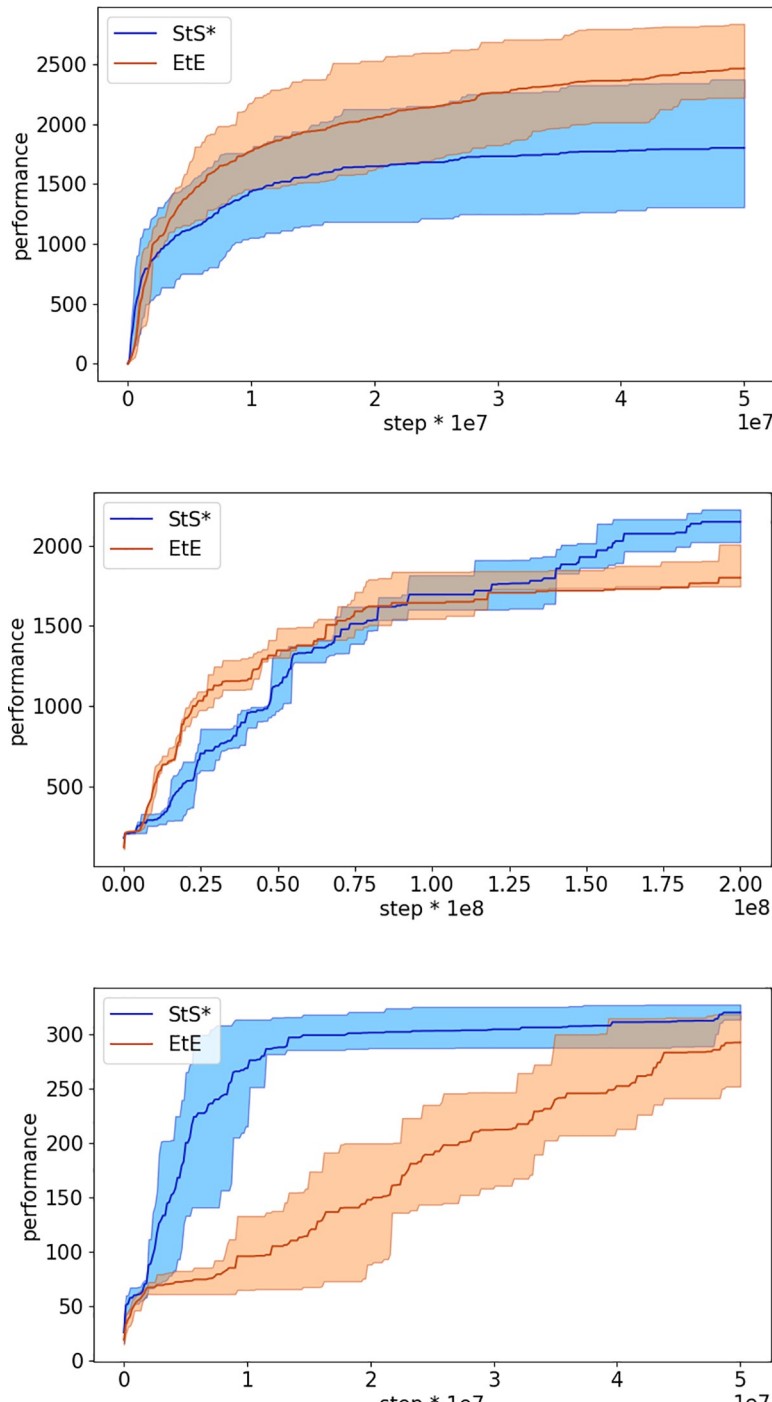

**Fig 5. Performance of the sequence to sequence condition in the continuous training modality (StS\*) and of end-to-end condition (EtE) during the evolutionary process in the case of the HalfCheetahBullet, BipedalWalkerHardcore, and MIT racecar problems (top, middle and bottom figures, respectively).** Performance refer to the average fitness of the best agent achieved so far. Data computed by post-evaluating the best individual of each generation for 3 episodes in the case of MIT racecar and HalfCheetahBullet and for 20 episodes in the case of the BipedalWalkerHardcore. Mean and 90% bootstrapped confidence intervals of the mean (shadow area) across 10 replications per experiment.

## Author Contributions

**Conceptualization:** Nicola Milano, Stefano Nolfi.

**Formal analysis:** Nicola Milano.

**Software:** Nicola Milano.

**Supervision:** Stefano Nolfi.

**Validation:** Stefano Nolfi.

**Writing – original draft:** Nicola Milano, Stefano Nolfi.

**Writing – review & editing:** Nicola Milano.

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
