## [Decision Letter · Decision Letter 0]

3 Mar 2021

PONE-D-21-03215

Autonomous Learning of Features for Control Experiments with Embodied and Situated Agents

PLOS ONE

Dear Dr. Milano,

Thank you for submitting your manuscript to PLOS ONE. After careful consideration, we feel that it has merit but does not fully meet PLOS ONE’s publication criteria as it currently stands. Therefore, we invite you to submit a revised version of the manuscript that addresses the points raised during the review process.

We look forward to receiving your revised manuscript.

Kind regards,

Josh Bongard

Academic Editor

PLOS ONE

Journal Requirements:

"NO"

"NO"

Reviewers' comments:

Reviewer's Responses to Questions

**Comments to the Author**

1. Is the manuscript technically sound, and do the data support the conclusions?

Reviewer #1: Yes

Reviewer #2: Yes

2. Has the statistical analysis been performed appropriately and rigorously? 

Reviewer #1: Yes

Reviewer #2: Yes

3. Have the authors made all data underlying the findings in their manuscript fully available?

Reviewer #1: Yes

Reviewer #2: Yes

4. Is the manuscript presented in an intelligible fashion and written in standard English?

Reviewer #1: Yes

Reviewer #2: Yes

5. Review Comments to the Author

Reviewer #1: This paper describes a a method of simultaneously training a feature extracting network alongside a control network. They validate their results across several well known benchmark problems. Although I am not a domain expert the paper was quite well written and relatively easy to understand.

Reviewer #2: Overall comments:

1. 'Episodes last 1,000 steps in the case of the Walker2DBullet, HalfCheetahBullet, and BipedalWalkerHardcore problems, and 10,000 steps in the case of the MIT racecar'. It is not clear how this is derived, is it based on past research or experimentally determined? Detailed justification of why MIT needs more steps is expected.

2. 'Data computed by post-evaluating the best individual of each generation for 3 episodes.' , is a confusing statement that needs to be made clear, preferable with including a table to explain results on figure 3. Each episode terminated after 1000 steps, however it is not clear how 1e7 steps equate to the 3 episodes.

3. The variable 's' should be defined in Figure 2 as done with the rest of other variables for clarity.

4. Discussion of the results is too brief, expected some postulations and thorough discussion on why the sequence-to-sequence model outperforms the other feature extracting methods used in previous works.

5. Limitations of this research and future direction of the study is not presented in this article.

Corrections:

Several typos to be attended as given below:

1. Finally, we compare different feature extracting methods and we show that sequence-to-sequence learning outperform the alternative methods considered in previous studies.

2. These approaches are usually referred with the term end-to-end learning.

3. Feature learning is a general domain which aims to extract features that can be used to characterized data.

4. Recently, this area achieved remarkably results in the context of neural network learning for classification and regression problems.

5. Moreover, we consider experimental scenarios that does not involve high-dimensional observations and, consequently, do not benefit from dimensionality reduction.

6. We introduce a method that permits to continue the train the feature extracting network during

7. The results reported by the authors demonstrates that the usage of the auto-encoder network permits to speed-up learning with respect to an end-to-end approach.

8. Finally, in the fourth phase, the control network is trained to drive the car so to maximize the cumulative rewards.

9. These problems are also qualitatively different from previous studied for what concerns the size of the observation vector

10. This enable us to verify whether features extraction

11. both aspects permits to significantly improve the advantage

12. The observation includes the position and orientation of the of the robot

13. the reward function includes also an additional component

14. The control network, that determine the state of the actuators, is constituted by a feed forward neural network with an a layer

15. one of the most effective evolutionary method for

16. There are also numerous typos further down mainly omission of 's' on refer, indicate,method, show, case etc that needs to be attended to.

6. PLOS authors have the option to publish the peer review history of their article (what does this mean?). If published, this will include your full peer review and any attached files.

Reviewer #1: **Yes: **John Rieffel

Reviewer #2: No

---

## [Author Response · Author response to Decision Letter 0]

25 Mar 2021

Response to reviewers are attached in the specific file

---

## [Editor Report · Decision Letter 1]

30 Mar 2021

Autonomous Learning of Features for Control Experiments with Embodied and Situated Agents

PONE-D-21-03215R1

Dear Dr. Milano,

We’re pleased to inform you that your manuscript has been judged scientifically suitable for publication and will be formally accepted for publication once it meets all outstanding technical requirements.

Kind regards,

Josh Bongard

Academic Editor

PLOS ONE
---

## [Editor Report · Acceptance letter]

6 Apr 2021

PONE-D-21-03215R1 

Autonomous Learning of Features for Control: Experiments with Embodied and Situated Agents 

Dear Dr. Milano:

I'm pleased to inform you that your manuscript has been deemed suitable for publication in PLOS ONE. Congratulations! Your manuscript is now with our production department. 

Kind regards, 

on behalf of

Dr. Josh Bongard 

Academic Editor

PLOS ONE